# A Comprehensive Review of Artificial Intelligence (AI) Applications in Pulmonary Hypertension (PH)

**DOI:** 10.3390/medicina61010085

**Published:** 2025-01-07

**Authors:** Sogol Attaripour Esfahani, Nima Baba Ali, Juan M. Farina, Isabel G. Scalia, Milagros Pereyra, Mohammed Tiseer Abbas, Niloofar Javadi, Nadera N. Bismee, Fatmaelzahraa E. Abdelfattah, Kamal Awad, Omar H. Ibrahim, Hesham Sheashaa, Timothy Barry, Robert L. Scott, Chadi Ayoub, Reza Arsanjani

**Affiliations:** Department of Cardiovascular Medicine, Mayo Clinic, Phoenix, AZ 85054, USA

**Keywords:** artificial intelligence, pulmonary hypertension, machine learning, deep learning, echocardiography, computed tomography, cardiac MRI

## Abstract

*Background:* Pulmonary hypertension (PH) is a complex condition associated with significant morbidity and mortality. Traditional diagnostic and management approaches for PH often face limitations, leading to delays in diagnosis and potentially suboptimal treatment outcomes. Artificial intelligence (AI), encompassing machine learning (ML) and deep learning (DL) offers a transformative approach to PH care. *Materials and Methods:* We systematically searched PubMed, Scopus, and Web of Science for original studies on AI applications in PH, using predefined keywords. Out of more than 500 initial articles, 45 relevant studies were selected. Risk of bias was evaluated using PROBAST (Prediction model Risk of Bias Assessment Tool). *Results:* This review examines the potential applications of AI in PH, focusing on its role in enhancing diagnosis, disease classification, and prognostication. We discuss how AI-powered analysis of medical data can improve the accuracy and efficiency of detecting PH. Furthermore, we explore the potential of AI in risk stratification, leading to treatment optimization for PH. *Conclusions:* While acknowledging the existing challenges and limitations and the need for continued exploration and refinement of AI-driven tools, this review highlights the significant promise of AI in revolutionizing PH management to improve patient outcomes.

## 1. Introduction

From new stethoscopes to intelligent clinical trials, artificial intelligence (AI) is revolutionizing healthcare. It has been particularly useful for dealing with the increasing volume of medical data and building systems that learn from experience. Machine Learning (ML), a subset of AI, focuses on developing algorithms that improve through data-driven learning. While traditional AI relies on hard-coded rules, ML can learn patterns and make predictions from data without explicit programming. This encompasses Supervised Learning, which uses labeled datasets to classify or predict data; Unsupervised Learning, which identifies patterns or relationships within unlabeled datasets; and Reinforcement Learning, which optimizes behavior in an environment to maximize rewards, forming the foundation of decision-making [1]. Deep Learning (DL), a more advanced subfield of ML, simulates neural network activity and has proven particularly useful in medical imaging through Convolutional Neural Networks (CNNs) [2]. These models can analyze medical images, though they require extensive data, computational power, and labeling efforts. Transfer Learning can reduce these demands by applying pretrained CNN models to new tasks [3]. Emerging AI advancements continue to accelerate innovation and efficiency in healthcare. By leveraging the power of AI, clinicians can potentially enhance the accuracy of disease diagnosis, optimize treatment strategies, and improve patient outcomes. Recent studies have demonstrated its ability to create tailored solutions that cater to specific medical disciplines and individual patient needs [4].

Pulmonary hypertension (PH) is defined hemodynamically by a mean pulmonary artery pressure (mPAP) over 20 mmHg, leading to increased right ventricular (RV) strain and often RV failure [5]. The significant increase in 1-year mortality of these patients is partly due to diagnostic delays caused by nonspecific symptoms and low sensitivity of standard tests like imaging, ECG, and pulmonary function studies [6]. Most PH cases (90%) are linked to left heart disease (WHO Group 2) or lung disease and/or hypoxia (WHO Group 3). Less common forms, such as WHO Group 1 [pulmonary arterial hypertension (PAH)] and Group 4 (associated with chronic pulmonary artery obstruction), require special attention due to the availability of unique treatment options. Chronic thromboembolic PH (CTEPH), in particular, is often curable, necessitating pulmonary vascular imaging for thorough evaluation.

Right heart catheterization (RHC) is the gold standard for diagnosing PH. However, it is an invasive procedure that carries risks, and its use is limited by logistical and financial burdens, as it often requires travel to specialized centers. In certain PH groups (e.g., WHO Groups 2 and 3), RHC provides limited diagnostic value, especially when PH-targeted therapies are not indicated [5], making non-invasive diagnostic tests and their interpretation more valuable. In the context of PH, AI can be employed to analyze vast troves of patient data, such as medical histories, symptom profiles, diagnostic test results, and imaging data, to uncover patterns and insights that may elude human clinicians [7]. Aiming to make non-invasive diagnostic tests more efficient, many studies have developed ML models, where algorithms analyze input and output data to learn how to predict the outcome of interest from the provided input. These models are then validated to evaluate their performance, both on the same dataset used for training (internal validation) and on independent datasets to assess generalizability (external validation). This comprehensive review article aims to explore the current and potential role of AI used with a variety of patient data in PH, drawing upon the latest research and evidence in the field.

## 2. Materials and Methods

This review was conducted by systematically searching for original studies that explored the applications of AI in PH. The databases searched included PubMed, Scopus, and Web of Science. The search strategy used combinations of key terms “artificial intelligence”, “machine learning”, “deep learning”, and “pulmonary hypertension”. Studies were included if they were original research articles focusing on AI applications in different aspects of PH. From an initial pool of over 500 articles identified during the search, 45 were selected based on their relevance. Non-original studies, review articles, editorials, and studies focusing on non-AI-based approaches were excluded, as well as duplicate articles. Snowballing techniques were not employed.

To assess the risk of bias in the included studies, we used the PROBAST (Prediction model Risk of Bias Assessment Tool). A summary of the PROBAST scoring for all included studies, detailing their risk of bias across key domains, is provided in Appendix A (Table A1). The quality of this narrative review was assessed using the SANRA scale, with detailed scores provided in Appendix A (Table A2).

## 3. Results

AI has been evaluated at multiple levels and across diverse data sources in various diseases. These include but are not limited to extracting and analyzing the demographic, clinical, and diagnostic data, as well as bioinformatics approaches for analyzing omics data to identify patterns, uncover biomarkers, predict disease risk, stratify patients, and guide therapeutic development. Specifically in PH, studies evaluated AI models using data from imaging modalities, ECGs, and phonocardiograms (Figure 1). 

### 3.1. Heart Auscultation

Auscultation, a fundamental aspect of physical examinations, can play a role in screening for PH. Analyzing the acoustic profiles of heart sounds has been explored as a method to uncover insights into underlying hemodynamic changes and to estimate pulmonary artery pressure [8,9]. Elgendi et al. developed a speech-based algorithm to explore whether PAH exhibits distinct acoustic signatures across the four auscultation sites [10]. From the recorded heart sounds, key features such as the relative power of specific frequency bands, energy of sinusoidal formants, and entropy were extracted. Linear discriminant analysis and leave-one-out cross-validation were employed to differentiate PAH subjects. The study identified a unique acoustic signature associated with PAH, highlighting its potential application in non-invasive screening tools. Another group of researchers explored the use of AI for the auxiliary diagnosis of congenital heart disease (CHD) and CHD-associated pulmonary arterial hypertension (CHD-PAH) using heart sound recordings [11]. They collected heart sound recordings from 1892 subjects diagnosed with CHD, including a subset of data from 373 patients diagnosed with CHD-PAH. The researchers used an RNN to distinguish between normal heart sounds and those indicative of CHD and CHD-PAH. They did not specify the characteristics of the patients used for training and validation of the AI model specifically for CHD-PAH. The RNN model was tested on a subset of 88 individuals, including 44 CHD-PAH cases and 44 controls. For this test set, the RNN exhibited an accuracy of 90% in correctly identifying individuals as positive or negative for CHD-PAH. Another study proposed a method for automatically detecting six categories of heart conditions, including PH [12]. They applied advanced transfer learning models, such as ResNet101 and DenseNet201, to classify different conditions from phonocardiograms (PCG), leveraging pretrained networks. The models achieved an accuracy of 0.90 to 0.98 in differentiating various conditions, including PH. These AI-driven approaches offer a potential avenue for early detection of disease, with innovative technologies augmenting traditional clinical methods.

### 3.2. Electrocardiography (ECG)

While a normal ECG cannot exclude PH, specific ECG abnormalities may suggest its presence and provide prognostic insights. Common findings include P pulmonale, right axis deviation and RV hypertrophy, right bundle branch block, and RV strain patterns [5]. AI algorithms have been widely used in ECG evaluation for various purposes, such as arrhythmia detection, structural heart disease detection, cardiovascular risk prediction, and signal enhancement [13]. DL shows strong accuracy in processing complex ECG data, which can be used in detecting conditions like PH and predicting survival measures [14]. By identifying intricate patterns within ECG signals, ECG-AI can play a unique role in PH screening, using a readily available, ubiquitous, and inexpensive test.

In a study by Kwon et al. [15], data were collected from over 38,000 patients across two hospitals, divided into different datasets for training and validation of an ML algorithm to detect PH. The algorithm employed an ensemble model combining a deep neural network (DNN) based on demographic (age, sex, weight, height) and ECG features (such as heart rate, PR interval, and QTc) with a CNN trained on raw 12-lead ECG waveform data. In internal validation, the algorithm achieved an area under the curve (AUC) of 0.859 (95% CI: 0.0855–0.863), and in external validation, it reached 0.902 (95% CI: 0.900–0.905), indicating strong predictive performance across both centers. A single-lead version of the AI algorithm was also developed, which demonstrated AUCs of 0.824 and 0.832 in internal and external validation, respectively. This suggests that wearable devices with single-lead ECG capabilities could also be viable for PH screening. The AI algorithm’s sensitivity map revealed that it focused more on the R-wave in narrow QRS complexes, the P-wave in intermediate QRS complexes, and the T-wave in wide QRS complexes when detecting PH. Their model also demonstrated predictive capability for the development of PH in individuals initially diagnosed as non-PH based on echocardiography, with the high-risk group showing a significantly higher incidence of PH development during follow-up (31.5% vs. 5.9%, *p* < 0.001).

Liu et al. developed and assessed ten AI models for identifying patients with elevated pulmonary artery pressure (ePAP) and its prognostic implications related to mortality [16]. Data from a hospital-based ECG database were utilized. Group 1 included patients who underwent transthoracic echocardiography (TTE) within two weeks of their ECG, with TTE providing PAP estimates. These data were used for model training, testing, and cross-validation. The cross-validation procedure was unconventional, using the validation set to select the best-performing model among ten AI models. The study followed up on true negative cases (AI-predicted non-ePAP with confirmatory PAP < 50 mm Hg) and false positive cases (AI-predicted ePAP with contradictory PAP < 50 mm Hg) with subsequent TTE assessments to track the development of ePAP. The DL models achieved a consistent AUC for identifying ePAP (mean: 0.88). Consistency was also observed for sensitivity (81.0%, 95% CI: 77.6–84.4%), specificity (79.6%, 95% CI: 77.4–81.7%), and accuracy (79.7%, 95% CI: 77.8–81.5%). V1, V2, and V3 were identified as the most important leads for the AI model to predict ePAP, which can be consistent with the clinical practice of reviewing the right precordial leads for evidence of RV enlargement. In comparison, conventional ECG characteristics, even when integrated, demonstrated low sensitivity (34.1%) for detecting ePAP. The study found that patients categorized by AI as ePAP, despite having normal PAP (<50 mm Hg) on initial TTE, were 5.04 times more likely to develop ePAP during follow-up, with an adjusted hazard ratio (HR) of 3.74 (95% CI: 3.28–4.26) compared to AI-identified non-ePAP patients. Patients in Group 2 had a longer interval between ECG and initial TTE (mean interval of 2.6 years, IQR: 0.7–4.1 years), and their data were used for ancillary validation of the AI model for cardiovascular mortality prediction. In this group, AI-predicted ePAP patients had a 6.61-fold higher likelihood of having ePAP detected by TTE during follow-up (HR 4.32; 95% CI: 3.91–4.78). During a 6-year follow-up of Group 1, the AI model’s stratification of patients as ePAP was independently associated with significantly higher cardiovascular mortality (HR 3.69) and all-cause mortality (HR 2.34) compared to non-ePAP patients, even after adjusting for potential confounding factors. These findings were consistent across various comorbidities and replicated in Group 2, further validating the model’s predictive power for mortality outcomes.

Another research team, led by Aras et al. [17], also used DL to detect PH using ECG data. Their study aimed to evaluate the ability of a DL approach to identify not only PH in general but also clinically relevant subtypes. They trained their algorithm on a dataset of 5016 PH patients (based on RHC or echocardiography) and 19,454 controls. The algorithm achieved an AUC of 0.89 for detecting PH overall (sensitivity of 0.79 and specificity of 0.84) and showed good performance in identifying precapillary PH, PAH, and PH due to lung disease and/or hypoxia. In their main data test with a 20% PH prevalence, the positive predictive value (PPV) was 0.56, and the negative predictive value (NPV) was 0.94. When the AI model was applied to a dataset simulating a general population (PH prevalence of 1%), the PPV dropped significantly to 0.05, while the NPV improved to 1.00, indicating that while the algorithm is highly reliable in excluding PH for patients identified as PH-negative, its ability to correctly identify PH cases in low-prevalence populations is very limited. The study also assessed the AI model’s ability to predict PH using ECGs obtained before clinical diagnosis, showing its highest performance at the time of diagnosis but maintaining strong predictive accuracy (AUC ≥ 0.79) up to 2 years before diagnosis.

DuBrock et al. [18] developed a CNN called the Pulmonary Hypertension Early Detection Algorithm (PH-EDA) to screen for PH and trained it on a large dataset of PH patients and controls. This algorithm was designed to classify patients as “PH-likely” or “PH-unlikely” based on RHC or echocardiography. Using ECGs recorded within 1 month of echocardiography or RHC, the PH-EDA achieved impressive results with an AUC of 0.92 in the test set and 0.88 in the external validation set from two medical centers. The algorithm demonstrated the ability to detect PH 6 to 18 months prior to diagnosis (AUC of 0.86 and 0.81 at two centers). Remarkably, it maintained an AUC of at least 0.73 in internal and external validation, using ECGs recorded up to five years prior to diagnosis. They also compared the performance of the PH-EDA to physicians’ manual interpretation of ECGs and found that the algorithm was able to identify PH even when waveforms did not show visually detectable abnormalities. In a subset of patients presenting with dyspnea, the prevalence of PH increased to 6%, compared to 3% in the overall study population. In this subset, the PH-EDA achieved a PPV of 23%, compared to 13% in the overall diagnostic test set, highlighting its potential to improve early detection in patients with this common yet nonspecific symptom of PH.

PH was one of 15 cardiovascular diagnoses for which Kalmady et al. developed and validated AI-based ECG models [19]. Their DL model, which utilized ECG trace data along with age and sex, demonstrated good predictive performance for PH, achieving an AUC in the range of 80–90%, indicating its potential utility in diagnosing the condition. The authors compared this model with another DL model that used ECG traces only and with an XGB model (a gradient-boosted decision tree algorithm) that used ECG measurements, age, and sex. They found that the DL model with ECG trace, age, and sex outperformed other models for most diagnoses, including PH. The QRS complexes were identified as the most influential region for PH detection. The authors acknowledge that the prevalence of PH was low in their sample, which may have impacted the PPV of the model.

Some studies have compared two different modalities, evaluating whether one offers superior performance or if their combination enhances diagnostic accuracy and predictive strength. Liu et al. developed and validated a DL model (DLM) using ECG data to detect ePAP, defined as a pulmonary artery systolic pressure (PASP) over 50 mmHg, indicated on TTE [20]. They also applied their DLM to Chest X-rays (CXRs) and further developed an integrated AI model combining ECG and CXR data for the first time. The AUCs were as follows: 0.8261 with ECG (sensitivity 76.6%, specificity 74.5%, NPV 97.6%, and PPV only 18.8%) and 0.8644 with a combination of ECG and CXR (sensitivity 78.6%, specificity 79.2%, NPV 98%, and PPV 22.5%) in the internal dataset. In the external validation dataset, the AUCs for ePAP detection were 0.8348 with ECG (sensitivity 78.9%, specificity 73.3%, NPV 97.7%, and PPV 20%) and 0.8734 with the combination (sensitivity 81.6%, specificity 78.1%, NPV 98.1%, and PPV 23.8%). However, in specific subgroups, including patients aged ≥65 and those with chronic kidney disease (CKD), heart failure (HF), or atrial fibrillation, the AUC of the DLM predictions based on ECG, CXR, and their combination was significantly reduced in both internal and external validations, showcasing decreased performance and generalizability. The authors also evaluated the long-term outcomes of their AI models in patients without initial evidence of ePAP. Those labeled as having ePAP by the models had significantly higher risks of developing left ventricular dysfunction (LVD, EF ≤ 35%) and cardiovascular mortality over 8 years. The combined AI-ECG + CXR model showed the strongest associations, with an HR of 4.51 for LVD and 6.08 for mortality, highlighting its prognostic value.

### 3.3. Imaging

AI is revolutionizing medical imaging, offering unprecedented opportunities for early disease detection, diagnosis, and prognostication. AI algorithms, particularly DLM, identify subtle patterns and features imperceptible to the human eye. These algorithms can automate tasks such as segmentation (identifying the boundaries of relevant structures, a crucial step in quantifying volumes and function), extraction of subtle disease features, and prediction of clinical outcomes to help clinicians identify and risk-stratify patients and tailor treatment strategies.

#### 3.3.1. Chest X-Ray

CXRs can reveal abnormalities in patients with PH and may also show signs of underlying conditions, such as left heart disease (LHD) or lung disorders [21]. However, the features suggestive of PH on CXR have limited sensitivity and specificity [22]. As the most widely used imaging modality, CXR contributes to a high volume of studies, creating a considerable workload. Furthermore, several studies have highlighted challenges and potential inaccuracies associated with interpreting these images [23]. AI-powered analysis has enabled radiologists to extract information more efficiently from images, improving the detection and classification of diseases based on CXRs [24]. Recent studies assessed AI in detecting CXR abnormalities in different cardiovascular conditions, including PH. They have utilized various DL models to analyze CXR images and identify patterns suggestive of PH. Researchers have trained these models on large datasets of CXR images from patients with and without PH, enabling the AI algorithms to distinguish between the two groups with promising accuracy.

Disease detection/prediction

One study investigated the potential of applying AI to CXRs to identify patients with ePAP, which is crucial for the timely detection and treatment of PH and to predict their risk of HF hospitalization [25]. They retrospectively analyzed data from 900 consecutive patients with suspected PH who underwent CXR and RHC as the diagnostic gold standard. A CNN was trained to identify patients with ePAP (>20 mmHg). The AI model’s performance was compared with traditional CXR measurements and eyeball assessments of CXR images by physicians blinded to RHC data in a separate validation set of 90 patients. The AI algorithm outperformed both human visual assessment and traditional CXR measurements, achieving an AUC of 0.71 for detecting ePAP (vs. 0.63 and 0.60, respectively, *p* < 0.05 for all comparisons). An independent group of 55 patients with suspected PH who were referred for RHC was used as the external validation dataset. Using Gradient-weighted class activation mapping (Grad-CAM) analysis, the authors found that AI focused on the right upper lung and cardiac regions in the elevated PAP group, while in the normal PAP group, the focus was around both hilar points. Patients identified as having ePAP by the AI algorithm had a two-fold increased risk of heart failure hospitalization compared to those without AI-predicted PH.

The same researchers investigated the ability of a DLM to predict Exercise-Induced Pulmonary Hypertension (EIPH) in 6 min walk (6 MW) stress echocardiography [26]. They used a previously developed AI model with residual blocks for PH prediction, with CXR images in a high-risk group [patients with scleroderma (SSc) or mixed connective tissue disease (MCTD) with SSc features]. Of 52 patients who were diagnosed with EIPH by 6 MW stress echocardiography, 29 underwent RHC. The AI model achieved an AUC of 0.71 for predicting EIPH. The study found that the DLM could predict EIPH and provide additional predictive value when combined with clinical and echocardiographic parameters (age, sex, blood pressure, and mean PAP at rest). They also used Grad-CAM to visualize the areas of the CXR images that the model focused on, which primarily highlighted the cardiac area in patients with EIPH.

Another study utilized transfer learning and fine-tuned three pretrained CNNs for detecting abnormalities in frontal CXR suggestive of PH [27]. The authors used a dataset of 405 PH patients and 357 healthy controls based on PASP values measured by Doppler TTE (≥40 mmHg and <40 mmHg, respectively). The models achieved high AUCs in both internal and external tests, indicating a high degree of accuracy and generalizability in classifying PH from normal CXRs, and outperformed the manual classification by an experienced radiologist in key performance metrics such as sensitivity, NPV, and F1-score. Additionally, the authors evaluated the models for predicting PASP values through a regression approach. The mean absolute error (MAE) for PASP predictions was lower in the internal test compared to the external test, suggesting limitations in predicting exact PASP values reliably across varied datasets.

Han et al. investigated the potential of AI in diagnosing CHD and PAH-CHD using frontal preoperative CXR images [28]. PAH-CHD status was determined by echocardiography. The researchers trained and evaluated the AI model using a dataset of 3255 frontal preoperative chest radiographs (1174 CHD radiographs, including 142 PAH-CHD and 2081 non-CHD radiographs). Due to the unbalanced numbers of samples in the PAH-CHD identification task, the PAH-CHD images were sampled at a ratio of 0.3 in the training set. The model achieved high accuracy in both CHD and PAH-CHD diagnosis (AUC 0.778, sensitivity 0.632, specificity 0.925, and accuracy 0.891 for identifying PAH-CHD). Notably, the study also conducted a multireader multicase (MRMC) study with five radiologists to evaluate AI assistance in diagnosis. A subset of 165 CHD [including 19 (11.52%) PAH-CHD] and 165 non-CHD radiographs were randomly selected, and radiologists diagnosed PAH-CHD from the CHD chest radiographs in two sessions, with AI-based classification provided for 50% of the images in each session. Diagnostic results were compared between radiologists and AI models, as well as radiologists with and without AI assistance. The results showed that AI achieved higher AUCs than manual identification of CHD and was comparable in PAH-CHD. No statistically significant difference was observed between radiologists without and with AI assistance in terms of sensitivity and specificity; however, AI assistance significantly improved the radiologists’ diagnostic accuracy (AUC increased from 0.639 to 0.705, *p* < 0.05).

With another DL algorithm to detect PAH using CXR images [29], researchers used a dataset of 519 CXR images, comprising 259 images from 145 patients with PAH diagnosed by RHC and 260 images from 260 control patients without suspicion of PH. The model, pretrained on ImageNet-1K, was used for image classification. The data sets were divided into training (418 images) and test sets (101 images), and a four-fold cross-validation was performed on the training set. At an optimal threshold, the algorithm achieved a sensitivity of 0.933 and a specificity of 0.982. GRAD-CAM analysis demonstrated that the model focused mostly on the pericardial areas of the CXR images. The AI model’s diagnostic performance in differentiating PAH from controls was superior to a group of nine experienced respirologists and radiologists (AUC of 0.988 and 0.945, respectively, *p* = 0.0175).

As mentioned earlier, Liu et al. developed a DLM to detect ePAP using CXR images [20], ECG data, and a combination of both. With CXR images alone, the model achieved an AUC of 0.8525 in internal validation (sensitivity 82.8%, specificity 72.7%, NPV 98.2%, and PPV 19%) and 0.8605 in external validation (sensitivity 85.4%, specificity 71.8%, NPV 98.3%, and PPV 20.2%).

Classification

In addition to disease detection, one study demonstrated that AI models can differentiate between different types of PH using CXR images [30]. CXR images from 2005 consecutive patients that had experienced signs/symptoms of PH, along with 2145 images from a publicly available dataset representing the healthy group, were included. Four physicians, blinded to patients’ medical history, visually assessed the CXRs for radiographic findings of PH based on established guidelines. Using a transfer learning approach, eight CNNs pretrained on ImageNet-1K were employed. In total, 6642 CXR images were categorized into six groups: five PH subtypes (PAH, PH due to LHD, PH due to lung disease and/or hypoxia, CTEPH, and PH with unclear and multi-factorial mechanisms), and a non-PH group. The best-performing proposed model achieved an accuracy of 86.14% in classifying the six categories, and ROC curve analysis showed an average AUC of 0.945.

#### 3.3.2. Echocardiography

Echocardiography is the recommended initial imaging modality for evaluating suspected PH [5]. It provides an estimate of hemodynamic parameters such as systolic pulmonary artery pressure (sPAP), which helps categorize patients based on their likelihood of having PH [31]. Despite its advantages, echocardiography has notable limitations. For instance, image quality may be compromised in patients with suboptimal acoustic windows, reproducibility can be inconsistent, and sPAP may be underestimated in certain patient populations [32]. Advancements in AI have the potential to address some limitations, making echocardiography a more reliable and powerful tool for screening and evaluating patients with suspected PH.

Automated Segmentation

Segmentation plays a crucial role in medical imaging by enabling the extraction of relevant qualitative and quantitative information. Deep learning architectures have shown success in cardiac image analysis, though they often focus solely on the left ventricle (LV). To enhance the segmentation performance of echocardiographic systems for early detection of right heart-related diseases, Cervantes-Guzman et al. developed a new segmentation pipeline based on the U-Net architecture [33]. They refined the model by incorporating pre- and post-processing steps, including heuristic corrections, to improve segmentation accuracy and align with clinical standards. Their method includes a cone segmentation module to isolate the relevant region of interest and a heuristic correction step to refine segmentation contours and address errors like overlapping chambers. The system was trained and validated on a dataset of echocardiographic images, including publicly available EchoNet-Dynamic data and clinician-annotated samples. Results showed a 2% improvement in the Dice score for overall segmentation accuracy after heuristic corrections, with better performance in identifying right heart chambers, which are typically more challenging. The system demonstrated high accuracy and consistency compared to manual annotations, highlighting its potential for aiding in PH diagnosis.

Disease detection

In a study in 2018, researchers developed a fully automated echocardiogram interpretation pipeline utilizing CNNs to analyze the cardiac structure and function and detect specific diseases, including PAH [34]. Using over 14,000 echocardiograms collected over a decade, models were trained for tasks like view classification, image segmentation, and disease detection. Researchers identified three diseases (amyloidosis, PAH, and chemotherapy-induced cardiotoxicity) in the main cohort and trained separate networks for each, using three random images per video for training. Specifically for PAH, CNN models analyzed the apical four-chamber (A4c) views of 584 studies of 104 patients and 2487 studies of 2180 controls. Accuracy was assessed using internal 5-fold cross-validation, and the model achieved an AUC of 0.85 (95% CI, 0.83–0.86), demonstrating promising diagnostic accuracy. While showing potential to facilitate access to echocardiographic diagnostics, limitations included challenges in analyzing low-quality images and complex cases like congenital anomalies, requiring further model refinement and validation.

Another study aimed to improve the diagnostic accuracy of PH by integrating ML with echocardiographic imaging, including 346 patients with suspected PH who underwent both echocardiography and RHC [35]. Among this cohort, 240 patients had PH. Data from 275 patients were used for training and internal validation (8:2 ratio), and 71 patients for external validation. Echocardiographic images of the parasternal short-axis papillary muscle level (PSAX-PML) view were preprocessed to extract features such as the long- and short-axis ratios and ventricular area measurements. The ML model outperformed traditional echocardiographic assessments in differentiating PH patients from controls, achieving an AUC of 0.945 (95% CI: 0.917–0.974) in the internal validation (vs. 0.892, *p* = 0.027) and 0.950 in the external validation (95% CI: 0.897−1.000). The model performed with similar accuracy in patients diagnosed with CHDPH and non-CHD–PH.

Another recent study explored a novel DL framework for automated detection and severity classification of PH in newborns using spatio-temporal patterns in TTE videos [36]. They utilized a total of 1311 echocardiograms from 270 newborns and explored two main approaches: a spatial approach, where manually curated frames were extracted from the TTE videos, and a spatio-temporal approach, which utilized short video sequences. Model performance was evaluated using a 10-fold cross-validation and a held-out test set. The researchers found that their spatio-temporal multi-view DLM, which incorporated information from five standard TTE views, achieved the highest performance on the validation set, with an F1-score of 0.84 for severity prediction and 0.92 for binary PH detection. The PSAX-PML emerged as the most informative single view for both tasks. However, on the held-out test set, the model performance decreased, with an F1 score of 0.63 for severity prediction and 0.78 for binary PH detection. The authors attributed this performance difference to the limited size of the test set and the inherent challenge of severity classification, suggesting that further model refinement with additional data may be beneficial. The inclusion of Grad-CAM saliency maps demonstrated that the model focused on clinically relevant cardiac structures like the interventricular septum and LV deformation, enhancing its explainability.

Anand et al. evaluated different ML models for predicting PH based on clinical and echocardiographic data [37], using data from 7853 patients who underwent RHC and TTE within one week. Patients were randomly split into 80% for training and 20% for testing. The training group was further divided into derivation (80%) and validation (20%) cohorts. Clinical and echocardiographic variables (features) that were commonly measured across the study population were included. XGBoost was chosen as the final algorithm due to its superior performance and ability to handle missing data. The final model included 19 features, achieving an AUC of 0.83 in the testing cohort, with a sensitivity of 88% and specificity of 54%. Importantly, the model performed well even when tricuspid regurgitation velocity (TRV, a key echocardiographic parameter for estimating PAP) was unavailable, achieving an AUC of 0.785. Key features driving the model’s performance included estimated right atrial (RA) pressure, atrial fibrillation/flutter on ECG, RV function impressions, body mass index, and heart rate. One of the limitations was the high prevalence of PH in the study population (81%), which may limit generalizability to broader screening settings.

Classification

A team of researchers investigated the use of ML models to enhance the classification of patients into non-PH, precapillary PH, and postcapillary PH categories [38]. They analyzed data from 885 patients who underwent both echocardiography and RHC. Using 24 echocardiographic and clinical parameters, the study trained and validated four ML models, selecting logistic regression with elastic net regularization as the best-performing classifier. Their model achieved AUCs of 0.789, 0.766, and 0.742 for non-PH, precapillary PH, and postcapillary PH, respectively. It demonstrated significantly higher accuracy than guideline-based echocardiographic assessments in the derivation cohort (59.4% vs. 51.6%, *p* < 0.01), and in the independent validation data set, its accuracy was comparable to the guideline-based PH classification (59.4% vs. 57.8%, *p* = 0.638). Key predictors included TRV for precapillary PH and left atrial volume index (LAVi), E/A ratio, and tissue Doppler imaging for postcapillary PH. The ML model faced difficulty in accurately classifying patients with combined precapillary and postcapillary PH, with 43% misclassified as precapillary PH compared to 16% in the non-combined group (*p* < 0.01), likely due to overlapping clinical and hemodynamic characteristics in this subgroup.

Prognostication

In a study, researchers investigated the prognostic value of an ML-based three-dimensional echocardiography (3DE) algorithm in assessing RV function in 151 patients with CTEPH [39]. The primary aim was to evaluate whether ML-derived RV ejection fraction (RVEF) could predict adverse clinical outcomes, including hospitalization due to right heart failure, interventions like pulmonary angioplasty, or death. Sixty-two patients underwent cardiac magnetic resonance (CMR) examination as well to assess RV volume and RVEF. The ML-based 3DE approach quantified RV structural and functional parameters, including RVEF, with high accuracy, correlating well with CMR measurements (RVEF correlation coefficient 0.849). Patients with adverse outcomes during the follow-up (median 19.7 months, interquartile range 0.5–54) had significantly lower RVEF (<30.3%), which independently predicted these outcomes with high sensitivity (98.7%) and moderate specificity (61.1%). Moreover, RVEF was closely associated with other prognostic indicators such as NT-proBNP levels, 6 MW distance, and cardiac index.

In another study, Fortmeier et al. aimed to refine the echocardiographic assessment of PH in patients with severe tricuspid regurgitation (TR), a condition where non-invasive measurements often underestimate pulmonary pressures [40]. A cohort of 116 patients with severe tricuspid regurgitation (TR) undergoing transcatheter tricuspid valve intervention (TTVI) were included in the study. The validation cohort consisted of 142 patients who also underwent TTVI for severe TR. Researchers developed an algorithm trained on echocardiographic parameters [left ventricular ejection fraction (LVEF), LV end-systolic diameter, left atrial (LA) area, sPAP, basal RV diameter, tricuspid annular plane systolic excursion (TAPSE), TR vena contracta width (VCW), RA area, and inferior vena cava diameter], to predict mPAP. While echocardiography identified only 27.2% of patients with elevated sPAP (>50 mm Hg), the AI model significantly improved the prediction of mPAP (R = 0.96, *p* < 0.0001). Notably, sPAP, TR VCW, LA area, and TAPSE showed the highest global feature importance for mPAP prediction and also stratified patients into prognostic groups based on predicted mPAP levels. Higher levels of predicted mPAP (≥29.9 mmHg) were associated with reduced 2-year survival following transcatheter tricuspid valve intervention (TTVI). This approach demonstrated potential for improving pre-interventional risk stratification while minimizing the need for invasive RHC.

RVEF is a critical prognostic marker in precapillary PH. CMR is the gold standard for its assessment but has limitations such as high costs and limited availability. Researchers aimed to use DL-based tools to bridge the gap between CMR and the most frequently performed cardiac imaging tool for RV function evaluation, two-dimensional (2D) echocardiography [41]. They developed a fully automated model to estimate RVEF from 2D echocardiographic images in patients with precapillary PH. This retrospective analysis included patients with suspected or confirmed precapillary PH who underwent echocardiography and CMR within one week. The DL model was trained to predict CMR-derived RVEF using a regression-based CNN and was assessed using five-fold cross-validation (training/test = 8:2). The DL model demonstrated a mean absolute error of 7.67% for RVEF prediction and a significant correlation with MRI-derived RVEF (r = 0.63, *p* < 0.001). It achieved an AUC of 0.84 for detecting severe RV systolic dysfunction (RVEF < 37%), comparable to RV fractional area change (FAC) measured by expert sonographers. The DL model also outperformed TAPSE in detecting mild RV systolic dysfunction (RVEF ≤ 45%), with an AUC of 0.87 compared to 0.67.

#### 3.3.3. Computed Tomography (CT)

CT plays a significant role in the diagnosis and management of patients with suspected or confirmed PH. Characteristic features include dilated pulmonary arteries, an enlarged right ventricle (RV) and atrium, and an elevated pulmonary artery-to-aortic ratio [31]. Beyond diagnosis, CT is a valuable tool for determining the underlying cause of PH and differentiating between various forms of PH. CT pulmonary angiography (CTPA) is used in the diagnosis of CTEPH, allowing visualization of clot distribution and assessment of suitability for available treatments [42]. Furthermore, CT aids in prognostic assessment by detecting signs of cardiac decompensation. While not the primary method for monitoring treatment response, it can provide information on changes in pulmonary artery size and RV function following therapeutic interventions. More recent CT techniques, including dual-energy CT or iodine mapping, offer promising avenues for the simultaneous assessment of pulmonary arteries and lung perfusion, potentially enhancing the diagnostic accuracy of CTEPH [32,42]. One of the limitations of traditional CT assessment in PH is its semi-qualitative nature, relying heavily on visual interpretation. Quantitative CT (QCT) involves extracting quantitative data from imaging, ranging from manual measurements of anatomical structures to advanced AI-driven texture analyses of lung parenchymal patterns. Modern QCT applications utilize ML and DL for automated, end-to-end analysis of CT scans, incorporating segmentation, quantification, and integration into clinical models. This holds the potential for leveraging vast amounts of data to improve early diagnosis, disease phenotyping, risk stratification, and treatment assessment [43].

Diagnosis and assessment

AI-powered segmentation tools can be used for precise mapping and analysis of pulmonary arteries and veins, facilitating early detection of vascular abnormalities and quantification of disease severity in PH.

Nardelli et al. developed an automated method for classifying lung vessels in chest CT images as arteries or veins, demonstrating its potential for precise vascular analysis [44]. The model was trained on the upper and lower lobes of the right lung from three non-contrast chest CT scans and evaluated on 18 additional cases from the COPDGene study, using manual classification as the reference standard. Their approach, which combined a three-dimensional (3D) CNN with graph-cuts optimization, achieved an overall accuracy of 93.6%, with sensitivity and specificity of 97% and 89%, respectively, outperforming alternative CNN architectures and random forest models. The model was further validated on 33 contrast-enhanced CT scans from patients with suspected CTEPH, including 18 cases confirmed by a panel of experts based on hemodynamic data and imaging characteristics. In this cohort, the model achieved an overall accuracy of 89.1%, with 86.9% for patients with CTEPH and 91.7% for controls.

In a more recent study [45], researchers investigated arterial and venous manifestations of vascular pruning and tortuosity in PAH and exercise pulmonary hypertension (ePH), not confounded by lung or thromboembolic disease, to quantify these morphologic changes through automated techniques. They included 42 patients with PAH, 12 with ePH, and 37 control subjects, all with available CT angiography (CTA). Pulmonary vasculature was reconstructed from the CT scans and labeled. The AI method enabled automated separation of arteries and veins, quantification of vascular volumes, and assessment of tortuosity in the pulmonary vasculature. Key findings include reduced small vessel volume and increased large vessel volume in the PAH group, along with higher arterial tortuosity compared to controls. In patients with ePH, similar pruning of small vessels was observed, suggesting early vascular remodeling. These AI-derived metrics demonstrate the potential for non-invasive identification of vascular changes in PAH and ePH, providing insights into disease progression and enabling quantitative phenotyping.

Melzig et al. investigated the use of automated 3D volumetry of central pulmonary arteries from CTPA images, combined with echocardiographic data, to predict mPAP and diagnose PH [46]. The study included 70 patients who underwent RHC, CTPA, and TTE for suspected PH. A stepwise multivariate linear regression model combining main pulmonary artery (MPA) volume from CTPA and echocardiographic PASP provided the highest diagnostic accuracy for predicting mPAP while excluding the axial diameter of the MPA. This model showed a strong correlation with invasive mPAP (r = 0.89, r² = 0.80, *p* < 0.001) and achieved an AUC of 0.94, outperforming models using the axial diameter of the MPA (AUC: 0.86), MPA volume alone (AUC: 0.90), or echocardiographic PASP alone (AUC: 0.92). A predicted mPAP > 25.8 mmHg identified PH with 86% sensitivity, 93% specificity, 95% PPV, and 81% NPV.

Zhang et al. developed a fully automated framework for diagnosing PH using CTPA images [47]. Eight pulmonary and cardiac substructures were segmented using a DLM. Morphological features from these segments were selected based on their relevance to pulmonary pressures while excluding overlapping or less significant features. An ML workflow incorporating XGBoost was applied to construct regression models for predicting mean, systolic, and diastolic pulmonary artery pressures (dPAP). Classification models were also built to stratify patients based on mPAP (cut-off 40 mmHg) and sPAP (cut-off 55 mmHg). A 10-fold cross-validation strategy was used to validate model performance. The segmentation model achieved high accuracy, with an average Dice score of 88.2%. The regression model using XGBoost demonstrated strong agreement with RHC measurements, with intraclass correlation coefficients (ICCs) of 0.934 (mPAP), 0.903 (dPAP), and 0.981 (sPAP). For classification, the XGBoost model achieved an AUC of 0.911 for mPAP and 0.833 for sPAP.

The dRV/dLV ratio (right-to-left ventricular diameter ratio) is a key parameter in PH studies. Automated ML-based measurement of dRV/dLV can offer advantages over manual methods, such as minimizing inter-observer variability and providing faster, more consistent results. Charters et al. used an automated AI tool to segment the ventricles on CTPA images and calculate the dRV/dLV ratio in 202 patients with suspected PH [48]. PH diagnosis was confirmed with RHC as the reference standard. dRV/dLV analysis was performed using a pretrained CNN model. There was a very good correlation between automated and manual dRV/dLV diameter ratio measurements (ICC 0.878, 95% CI 0.837–0.908). Compared to PH-negative patients, patients with PH were significantly more likely to have an elevated automated dRV/dLV ratio. In the derivation cohort, the AUC for predicting PH was 0.752 (95% CI: 0.677–0.827, *p* < 0.001) for the automated dRV/dLV ratio and 0.615 (95% CI: 0.527–0.704) for the manually measured dRV/dLV ratio. Using a threshold dRV/dLV ratio of 1.12 for predicting PH, the method achieved optimized sensitivity and specificity (73% and 67%, respectively). The automated dRV/dLV ratio also outperformed the manual measurements of the MPA and the ratio of MPA to the ascending aorta diameter (MP/AAo) in PH screening. Among the 5 PH subtypes, the automated dRV/dLV ratio was highest in Group 1, significantly greater than in Groups 2 and 4, while Groups 3 and 5 showed lower ratios. Additionally, over a median follow-up of 300 days (range: 8–928, interquartile range: 139 days), a higher dRV/dLV ratio was strongly associated with increased mortality risk (HR 6.5, 95% CI: 2.5–16.82, *p* < 0.001).

CTEPH can be misdiagnosed due to its nonspecific symptoms and reliance on specialized imaging modalities like ventilation/perfusion (V/Q) scans and CTA. In a study, Zhao et al. created an automatic method for diagnosing CTEPH using non-contrasted computed tomography (NCCT) scans [49]. This approach eliminates the need for precise lesion annotation, which is a significant challenge in CTEPH diagnosis due to small lesion sizes and complex diagnostic procedures. Scans from 300 subjects (132 normal cases, 88 CTEPH cases, and 80 cases of other lung diseases) were used. A novel cascaded network with multiple instance learning (CNMIL) framework was developed. Multiple instance learning (MIL) uses attention scoring to pinpoint the most diagnostically important parts of the images. Five-fold cross-validation was used to test the model’s performance. The framework outperformed traditional and state-of-the-art diagnostic methods for CTEPH and achieved high diagnostic accuracy, with an AUC of 0.807, accuracy of 0.833, sensitivity of 0.795, and specificity of 0.849 in distinguishing CTEPH cases.

Gawlitza et al. investigated a supervised ML model that utilized quantitative and qualitative CTA features for the non-invasive prediction of hemodynamic outcomes in CTEPH patients [50]. The study included 127 patients with diagnosed CTEPH who underwent preoperative RHC and thoracic CTA. The researchers extracted 19 qualitative and quantitative imaging features along with three hemodynamic parameters: mPAP, RA pressure (RAP), and pulmonary artery oxygen saturation (PA SaO2). A random forest algorithm was employed to predict hemodynamic risk groups based on these features, with feature importance calculated to identify key predictors. Moderate to strong correlations were demonstrated between several imaging features and hemodynamic parameters. Key imaging features for mPAP prediction included the RA/LV ratio, mosaic attenuation, left PA diameter, and contrast retention in hepatic veins, while PA SaO2 prediction relied on contrast retention, MP/AAo ratio, pericardial effusion, and RA/LV ratio. The model achieved an AUC of 0.82 for predicting elevated mPAP and 0.74 for PA SaO2.

AI-powered tools can also identify and quantify lung disease patterns by integrating various radiomic techniques such as texture analysis and classification. A novel AI framework was developed by Mamalakis et al. using DL techniques with transparency and interpretability [51]. The framework analyzes 3D CT imaging models to classify different lung patterns in PH patients. Around 17,500 CT slices were divided into six classes (healthy lungs, ground glass, ground-glass reticulation, honeycomb, emphysema, and unhealthy lungs). These classes were manually labeled in each slice by two specialist radiologists. The model was trained and validated on an 84-patient “seen” cohort and externally tested on a 19-patient “unseen” cohort. Among the DL models tested, DenRes-131 achieved the best performance with a classification accuracy of 93.69%. To enhance transparency and reliability, the study combined Grad-CAM with Principal Component Analysis (PCA) to evaluate the AI model’s learning patterns and reduce bias in its decision-making. This approach, using PCA-GradCam for Grad-CAM outputs and PCA-Shape for input image variability, allowed the researchers to assess the consistency and accuracy of the model’s predictions. The study demonstrated the framework’s ability to identify emphysema and honeycomb patterns with high confidence, although it faced challenges with ground-glass reticulation and healthy patterns.

Prognostication

Another study focused on incorporating AI-quantified fibrosis into the assessment of patients with PH [52]. A total of 521 adult patients with idiopathic PAH (IPAH) or PH associated with lung disease (PH-LD) who underwent CT imaging were included. An established AI model was used to quantify fibrosis on CTPA images. Radiologists also scored fibrosis on the scans, defining fibrosis as a composite of ground-glass reticulation and honeycombing. The AI-derived fibrosis percentage was independently associated with increased mortality (HR: 1.01, *p* = 0.04) and showed strong predictive performance in an external validation cohort (C-index: 0.76). A fibrosis threshold of 3.43% corresponded to 20% 1-year mortality, with significantly poorer survival for patients exceeding this threshold (1-year survival: 69% vs. 88%, *p* = 0.001). The combination of AI-quantified fibrosis and radiologic severity scoring improved the accuracy of survival prediction compared to visual scoring alone (C-index: 0.67 vs. 0.61, *p* < 0.001). Notably, the AI model detected subtle fibrosis in patients visually scored as having no fibrosis.

Shikhare et al. explored the potential of the dRV/dLV ratio measured using ML algorithms on CTPA images [53] to predict outcomes in CTEPH patients undergoing pulmonary endarterectomy (PEA). The study included 99 CTEPH patients, with an ML algorithm applied to segment the ventricles and calculate the dRV/dLV ratio from preoperative CTPA images. These measurements were compared to manually calculated dRV/dLV ratios, and their association with postoperative outcomes was analyzed using multivariable linear regression. The ML algorithm successfully segmented the ventricles in 79% of cases, and ML-derived values demonstrated a strong correlation with manual measurements (r = 0.9, *p* < 0.001). At preoperative RHC, the dRV/dLV ratio showed moderate correlations with mPAP (r = 0.6, *p* < 0.0001) and pulmonary vascular resistance (PVR) (r = 0.5, *p* < 0.0001). A dRV/dLV ratio ≥ 1.2 was associated with worse hemodynamics, including lower cardiac output, higher mPAP, and higher PVR (*p* < 0.001). At surgery, these patients required longer cardiopulmonary bypass (*p* = 0.037) and circulatory arrest times (*p* < 0.001) and were more likely to have proximal disease (Jamieson Type 1/2, *p* = 0.032). Post-operatively, a dRV/dLV ratio ≥ 1.2 was associated with longer ICU and hospital stays, while a ratio ≥ 1.6 was strongly linked to the need for ECMO (Extracorporeal Membrane Oxygenation) (*p* = 0.006). The ML tool failed in 21% of cases, predominantly in patients with severe PH and poorer functional status, which was linked to suboptimal contrast opacification in CT images. Logistic regression analysis for predicting ICU and hospital length of stay showed AUCs of 0.76 and 0.79, respectively, highlighting the potential utility of dRV/dLV in risk stratification.

#### 3.3.4. MRI

CMR is a powerful, non-invasive imaging modality that can play a vital role in the diagnosis, prognosis, and management of PH. It can provide detailed anatomical and functional insights, assess pulmonary arterial stiffness (PAS), differentiate between precapillary and combined pre- and postcapillary PH using parameters such as the interventricular septal angle, and monitor treatment response by tracking changes in key metrics over time. While CMR offers a wealth of information, the conventional analysis of images requires significant expertise and time. Several studies have already demonstrated the potential of AI in CMR analysis for PH.

Automated Segmentation and Feature Extraction

One study introduced a deep learning-based pipeline for fully automatic 3D bi-ventricular segmentation of CMR images, designed to handle challenges including imaging artifacts and complex cases such as PH [54]. The pipeline combined a multi-task network for segmentation and landmark localization with shape refinement using anatomical priors, ensuring smooth and clinically meaningful results. It retained 3D spatial consistency while maintaining computational efficiency through a 2.5D representation. The pipeline’s ability to produce accurate segmentation was validated and achieved Dice scores of 0.946 ± 0.026 for the LV cavity and 0.912 ± 0.041 for the RV cavity on a dataset of 1831 high-resolution volumes, and 0.926 ± 0.031 for the LV and 0.892 ± 0.046 for the RV on 600 simulated low-resolution volumes. While the pipeline achieved high overall accuracy, the accuracy for the RV wall was relatively lower. To experiment with the model on pathological low-resolution volumes, it was trained and tested on the PH subgroup of 20 patients. It effectively segmented images with abnormal RV dilation and deformation, enabling quantification of RV structure and function. The RV cavity volume was measured as 146.69 ± 52.96 mL using the model, compared to 148.73 ± 54.23 mL from manual segmentation of the corresponding high-resolution images.

A group of researchers studied metabolic profiles linked to myocardial adaptation under biomechanical stress in PH [55]. They combined 3D biomechanical modeling of the RV, using CMR, with statistical analysis to identify biomarkers predictive of survival and adverse RV remodeling. Using automated segmentation, the researchers developed a detailed 3D map of RV geometry, including wall stress distribution, and linked this to 10 biomarkers previously associated with PH survival [for instance, dehydroepiandrosterone sulfate (DHEA-S)]. Increased wall stress was strongly correlated with adverse RV remodeling and survival outcomes. Wall stress independently predicted all-cause mortality (HR 1.27, *p* = 0.04). This integrative approach demonstrates the interplay between mechanical stress and metabolic pathways, providing insights into RV maladaptation and identifying potential therapeutic targets.

As part of their study, Ma et al. applied a DL-based approach for automated RV segmentation in CMR to improve the assessment of PH [56]. The YOLO (You Only Look Once) algorithm and CNN outperformed traditional methods like active shape models and random forests, achieving a Dice coefficient of 0.89 and an accuracy of 99.4%. The AI model identified key morphological features of PH, including RV anterior wall thickening and pulmonary artery dilation.

These models achieved precise and comprehensive RV segmentation without requiring additional imaging, addressing the complexities of CMR analysis and providing an accurate alternative to echocardiography for RV evaluation.

Diagnosis and Differentiation of PH

A study by Lungu et al. employed ML to enhance the non-invasive diagnosis of PH using MRI-derived computational models and imaging metrics [57]. The study included 72 patients who underwent both RHC and cardiac MRI. By integrating physiological metrics, such as PVR, compliance, and wave reflection ratio (Wb/Wtot, ratio of backward to total wave power), with anatomical features like RV ejection fraction, ventricular mass index, and systolic septal angle, the ML model achieved a diagnostic accuracy of 92% (sensitivity: 97%; specificity: 73%). The addition of these image- and computation-derived metrics significantly improved diagnostic performance, increasing the AUC from 0.89 for individual metrics to 0.91 when combined. The wave reflection ratio emerged as the strongest standalone indicator of PH, while parameters like pulmonary artery relative area change added complementary diagnostic value. This study highlights the potential of combining advanced imaging techniques with computational models to reduce reliance on invasive procedures like RHC in diagnosing PH.

In another study, a tensor-based ML approach was applied to CMR imaging for the diagnosis of PAH [58]. This method utilized multilinear subspace learning to reduce high-dimensional CMR data into interpretable diagnostic features without requiring manual segmentation. The model achieved a diagnostic accuracy of 0.90 for short-axis (SAX) cine images and 0.86 for four-chamber (4Ch) cine images. For differentiating idiopathic PAH (IPAH) from controls, the accuracy increased to 0.97 and 0.95, respectively. The study identified key features such as interventricular septal deviation and abnormal RV free wall contraction and discovered novel indicators, including patterns in the basal LV lateral wall. Temporal analysis revealed that features at end-systole provided the highest discriminative power. This approach also demonstrated efficiency, processing CMR data and delivering accurate diagnostic outputs in under 10 s.

A radiomics-based ML study evaluated texture features extracted from CMR to differentiate PH patients from healthy controls [59]. Using a dataset of 72 patients (42 PH and 30 controls), texture features were derived from the mid-LV myocardium via balanced steady-state free precession cine SAX imaging. Five-fold repeated cross-validation with five repeats was performed to evaluate the predictive performance of 12 models. The multilayer perceptron (MLP) model achieved the best performance, with an AUC of 0.862 and 78% accuracy in the primary analysis and an AUC of 0.918 and 80% accuracy in a subgroup analysis of PH patients with preserved LVEF (LVEF ≥ 50%). The study demonstrates the potential of radiomics to noninvasively detect PH, including in patients without systolic dysfunction.

The same team applied a radiomics-based ML approach to CMR for PH diagnosis, comparing multiple data augmentation and feature selection techniques [60]. Data from 82 subjects (42 PH and 40 controls) were analyzed. Using radiomic features extracted from LV and RV myocardial masks, the researchers introduced the Data Augmentation for Information Transfer (DAFIT) technique, which outperformed models trained on unprocessed data or using conventional feature-filtering approaches. DAFIT achieved the highest diagnostic accuracy, with an AUC of 0.958 for the combined LV-RV mask model.

Combined LV and RV masks performed better than individual masks, and model performance was consistent even for PH patients with preserved ejection fraction.

Wang et al. investigated the use of DL-based CMR for screening and diagnosis of cardiovascular diseases, including PAH [61]. A two-stage AI-enabled approach was developed using a vast dataset of 9719 patients, encompassing 11 types of cardiovascular diseases (including 200 PAH patients). The screening model, using cine CMR sequences, achieved an AUC of 0.988 for anomaly detection. For PAH diagnosis specifically, the diagnostic model, which combined cine and late gadolinium enhancement (LGE) imaging, attained an AUC of 0.998 and outperformed expert cardiologists in identifying cases, even those without visible CMR abnormalities. Advanced algorithms like video-based Swin Transformers improved motion analysis, while Grad-CAM visualizations highlighted clinically relevant features, such as RV abnormalities. The model demonstrated robust generalizability across diverse datasets, enabling precise quantification of RV function, pulmonary arterial stiffness, and vascular changes critical for risk stratification and treatment monitoring.

Prognostic Assessment and Risk Stratification

A supervised ML model was developed to predict survival in PH patients by analyzing 3D RV motion patterns derived from CMR [62]. The study included 256 patients with newly diagnosed PH, and the ML model identified key motion predictors of poor outcomes, such as reduced basal longitudinal motion and transverse contraction in the septum and RV free wall. Incorporating 3D motion into survival prediction improved the AUC from 0.60 to 0.73 (*p* < 0.001) and stratified high- and low-risk groups with a significant difference in median survival (13.8 vs. 10.7 years, *p* < 0.001).

Alandejani et al. developed and validated a DLM for automated RA area measurement using CMR in patients with PH [63]. Trained on a multicenter cohort of 365 subjects, the model demonstrated higher repeatability (ICC 0.91–0.95) than manual measurements (ICC 0.82–0.91). In a clinical testing cohort of 400 patients, the AI-derived RA area measurements maintained accuracy and showed moderate correlations with invasive hemodynamics (r = 0.64 for mean RAP). The AI model also effectively predicted the European Society of Cardiology/European Respiratory Society (ESC/ERS) risk thresholds for mean RAP, with AUCs ranging from 0.82 to 0.90. Furthermore, both AI and manual RA area measurements were equally predictive of mortality, with hazard ratios of 1.02 (*p* < 0.01).

Another DL model for automated CMR analysis was developed and validated using a multicenter dataset with diverse cardiopulmonary conditions, including PH [64]. The model demonstrated positive correlations with RHC parameters, such as mPAP (r = 0.56) and PVR (r = 0.62), outperforming manual measurements. Prognostic assessment in 3487 patients confirmed the utility of RV parameters, including RV ejection fraction (HR 0.80, *p* < 0.001), RV end-diastolic volume, and ventricular mass index, for predicting mortality in PH. The model showed high interstudy repeatability (ICC > 0.90) and generalizability across multiple vendors and centers, which can reduce variability in RV assessments.

In another study, authors utilized a multilinear principal component analysis (MPCA)-based ML model to extract prognostic features from CMR for predicting 1-year mortality in treatment-naïve PAH patients [65]. Using data from 723 patients, the model combined SAX and 4Ch CMR features, dynamically assessed across the cardiac cycle. Significant predictors included end-systolic interventricular septum flattening and end-diastolic LV changes, which reflect RV pressure overload and impaired LV filling. The MPCA model improved the 1-year mortality prediction of the REVEAL score (which predicts survival in patients with PAH based on clinical and echocardiographic variables) from a c-index of 0.71 to 0.83 (*p* < 0.001) when combined with CMR volumetric data. Kaplan–Meier survival analysis confirmed the utility of MPCA features in stratifying high- and low-risk groups, with the most impactful prognostic features observed in the septum during systole and LV during diastole.

These studies collectively highlight the diverse applications of AI in PH, encompassing screening, diagnosis, classification, and prognostication. A summary of all the original studies included in this review is provided in Table 1.

## 4. Discussion

AI has shown remarkable potential in addressing the diagnostic and prognostic challenges of PH. By leveraging data from various modalities, AI-driven tools are transforming PH care.

Advancing Diagnosis and disease classification

AI can enhance diagnostic accuracy through advanced imaging and data analysis. Models trained on imaging data detect subtle features like right heart strain and pulmonary artery enlargement, often surpassing manual methods. ECG-based algorithms effectively identify patterns indicative of ePAP. Similarly, phonocardiogram analysis reveals acoustic signatures associated with PH, providing non-invasive screening options, particularly in resource-limited settings.

Patients with PH are a heterogeneous group with varying underlying etiologies, such as PAH, left heart disease, and CTEPH. AI tools show promise in stratifying patients within these diverse groups, facilitating more personalized diagnostic and prognostic approaches.

Risk Stratification and Prognostic Applications

AI models show potential in risk stratifying and predicting outcomes such as mortality and hospitalization risk. ML algorithms, utilizing imaging features, can guide therapeutic decisions with greater precision than traditional approaches. These tools support personalized treatment planning and long-term management strategies.

Challenges and limitations

It is important to acknowledge certain limitations in the existing studies that evaluate AI models to be used in PH. The reliance on large, high-quality datasets for training AI models necessitates careful consideration of data bias and generalizability. Among the studies included in this review, some exhibited a high risk of bias due to issues such as limited validation, small sample sizes, or poorly defined outcomes. These factors may impact the reliability of the overall conclusions and necessitate cautious interpretation of the findings. The development and evaluation of diagnostic and prognostic models for PH face inherent challenges related to patient selection and diagnostic criteria. Studies are often retrospective and rely on a combination of RHC and echocardiographic data to label patients as having or not having PH. This approach introduces potential selection bias: patients undergoing RHC are typically higher-risk cases with a strong suspicion of PH, which may not represent the broader population where such models could have the greatest impact. Conversely, using echocardiographic criteria to include patients can increase sample size and generalizability but at the cost of incorporating cases with varying probabilities of PH rather than confirmed diagnoses. These methodological constraints highlight the need for careful consideration of population characteristics and diagnostic definitions in model development.

Future research should prioritize rigorous validation methods to improve model reliability and address common limitations such as imbalanced datasets, overfitting, and lack of external validation. Researchers should ensure robust reporting by clearly defining the intended use of AI models, whether for screening high-risk populations or aiding diagnosis, and aligning training and validation datasets accordingly. For example, datasets for screening models must include sufficient cases from the target population to ensure realistic performance evaluation. Key performance metrics such as sensitivity, specificity, PPV, and NPV should be reported and stratified by subpopulations to demonstrate clinical applicability. Studies should also address potential biases, such as imbalanced datasets or overrepresentation of low-risk or high-risk individuals, which may skew performance metrics. External validation on independent datasets that reflect their targeted populations is essential to confirm generalizability and ensure clinical utility.

Furthermore, the integration of AI models into routine clinical practice will require addressing barriers such as interoperability with electronic health record systems and ensuring clinicians’ trust through explainable AI models. The “black box” nature of some AI algorithms can hinder interpretability and clinical acceptance. Addressing this challenge through rigorous validation studies, transparent algorithm development, and ongoing research aiming for robust methodology is crucial for ensuring reliable and clinically meaningful results.

Future Directions

In addition to the established potentials of AI in early diagnosis, disease classification, and prognostication in PH, future advancements hold the promise of deeper transformations in how we understand and manage this complex condition. AI assists multimodality deep phenotyping, which can offer unprecedented insights into disease mechanisms and refine clinical classifications of PH. Advancements may also pave the way to transitioning from invasive procedures like RHC to non-invasive modalities for diagnosis and monitoring, reducing risks and improving accessibility. Furthermore, as AI evolves, its integration into routine clinical workflows could become seamless, combining automated analysis of imaging tests with clinical and laboratory data to enhance decision-making and personalized medicine. Future research should also focus on expanding AI applications in treatment and patient care, such as drug discovery or repurposing and remote patient monitoring (Figure 2). Additionally, end-stage PH patients (cardiogenic shock or advanced heart failure) may be in need of more sophisticated therapies such as mechanical circulatory support. The role of AI is being evaluated in the critical care field, holding an enormous potential for improving patient outcomes in this setting [66,67]. These developments, coupled with advancements in the interpretability and transparency of AI models, will ensure that these tools are not only effective but also trusted by clinicians and patients alike.

## 5. Conclusions

The findings highlighted in this review demonstrate the immense potential of AI to revolutionize the diagnosis, management, and prognostication of PH. PH remains a challenging condition to diagnose early, primarily due to its nonspecific symptoms and the inherent limitations of conventional diagnostic tools. AI, with its ability to process and learn from complex datasets, offers promising solutions to these challenges by enhancing the accuracy, efficiency, and scope of clinical assessments.

## Figures and Tables

**Figure 1 medicina-61-00085-f001:**
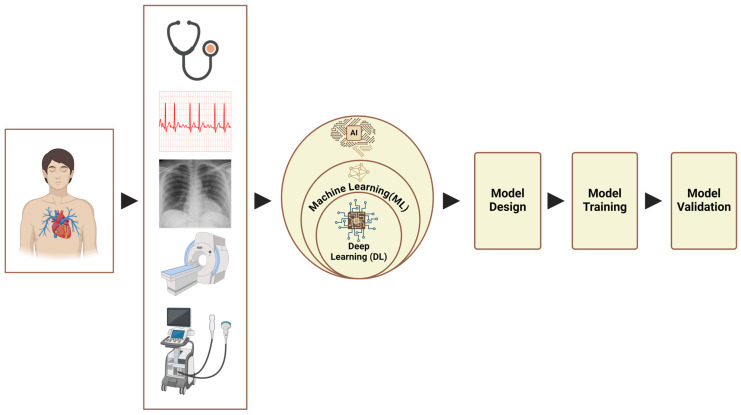
AI model development workflow for use in pulmonary hypertension.

**Figure 2 medicina-61-00085-f002:**
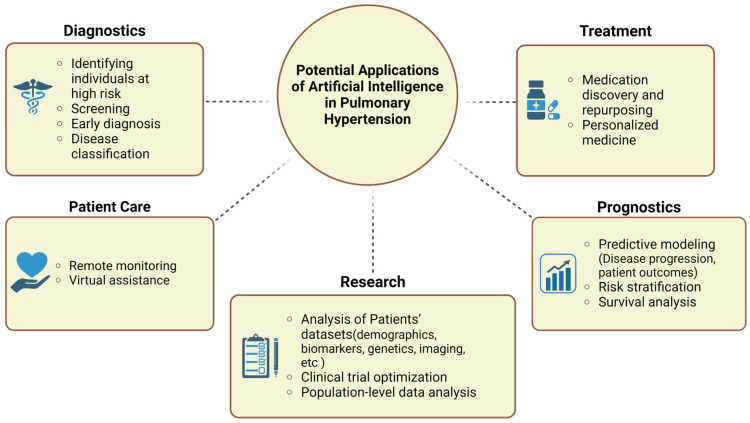
Potential applications of artificial intelligence in pulmonary hypertension.

**Table 1 medicina-61-00085-t001:** Summary of studies evaluating AI applications in pulmonary hypertension.

	Study	Study Population	Diagnosis	Key Findings	Limitations
**Auscultation**	Elgendi et al. [10] (2018)	60 subjects, including 25 PAH and 35 non-PAH.	RHC.	The model revealed a PAH vs. non-PAH classification sensitivity of 84% and specificity of 88.57% using heart sounds collected from 2nd left intercostal space.	-Limited generalizability.
	Yang et al. [11] (2023)	-Model used in a subset of 373 patients diagnosed with CHD-PAH -The study did not specify the details of the training and validation procedures for CHD-PAH diagnosis.	Echocardiography (confirmatory testing not performed for all individuals).	In a set of 88 individuals, including 44 CHD-PAH cases and 44 controls, the RNN exhibited an accuracy of 90% in correctly identifying individuals as positive or negative for CHD-PAH.	-Limited generalizability. -Insufficient data on the CHD-PAH subgroup. -Lack of confirmatory echocardiography for all screened individuals, potentially missing some cases.
	Wang et al. [12] (2022)	-PH group: 74 subjects (102 recordings initially; used 200 heart sound recordings) from a clinical setting. -Other classes of heart diseases from an open database with 1000 recordings.	Details not provided.	-Applied transfer learning architectures to classify data for training and testing the models in 10-fold cross validation (9:1 training to testing sets). -Some models (e.g., ResNet101, DenseNet201) achieved an accuracy of 98% in multi-class classification of six heart sound categories, including PH.	-Limited generalizability of PH group. -Lack of detailed reporting on PH diagnosis criteria.
**ECG**	Kwon et al. [15] (2020)	Hospital A (derivation: 56,670 ECGs from 24,202 patients, and internal validation: 3174 ECGs/patients). Hospital B (external validation: 10,865 ECGs/patients).	Echocardiography (parameters per 2015 ESC/ERS guidelines).	-Predicted PH with AUC 0.859 (internal) and 0.902 (external). -In a subgroup of 2939 patients, the group flagged as high risk had higher PH incidence compared to low-risk patients during 24 months of follow-up (31.5% vs. 5.9%, *p* < 0.001).	-PH is not diagnosed based on RHC. -Limited external validation (generalizability).
	C. Liu et al. [16] (2022)	Group 1: 41,097 patients (randomly assigned to 10 independent segments: 8 to the training, 1 to the validation, and 1 to the test set), including 10,818 patients followed up for developing ePAP with another TTE for risk prediction). Group 2: 18,373 patients. Data used for ancillary validation of the model for cardiovascular mortality. A different external validation cohort of 279 patients was also examined.	PASP > 50 mmHg estimated by TTE.	-Group 1: AUC 0.88 for ePAP detection. Among patients that had normal PAP estimated by TTE, AI-predicted ePAP patients had a 5.04-fold risk of developing ePAP (multivariate-adjusted HR: 3.74) -The 6-year cardiovascular mortality risk was higher in patients stratified as ePAP by the AI model compared to those stratified as non-ePAP (HR 3.69).	-ePAP estimated by TTE was selected as a surrogate marker for highly suspicious PH; no other marker was tested. -Limited external validation.
	Aras et al. [17] (2023)	5016 PH patients, 19,454 non-PH controls. Patients randomly divided into training, development, and test sets in a ratio of 7:1:2.	RHC or TTE (TRV > 3.4 m/s) if RHC unavailable.	Detected PH with an AUC of 0.89 (sensitivity of 0.79, specificity of 0.84) and subtypes with AUCs ≥ 0.79 up to 2 years prior to diagnosis.	-PH is not diagnosed based on RHC in all cases. -Patients could have been misclassified in the PH subtypes. -Limited external validation.
	DuBrock et al. [18] (2024)	Hospital A: 39,823 PH-likely and 219,404 controls [randomly split into training (48%), validation (12%) and test (40%) sets]. Hospital B: 6045 PH-likely and 24,256 controls as test set.	RHC or TTE (PH-likely: TRV > 3.4 m/s) if RHC unavailable (parameters per 2022 ESC/ERS guidelines).	PH-EDA detected PH up to 5 years prior to diagnosis, with AUCs ranging from 0.79 to 0.92 across datasets.	-PH is not diagnosed based on RHC in all cases. -Limited external validation.
	Kalmady et al. [19] (2024)	PH was one of 15 conditions in a cohort of patients presenting to 84 hospitals. In this subgroup, 36,331 ECGs (development set: 21,869 ECGs, holdout set: 14,462 ECGs) were used.	ICD-10 codes.	Predicted PH with AUC 84.77%.	-PH status was decided based on ICD codes. -Limited generalizability, unable to perform external validation (used a holdout cohort). -Broad dataset may dilute PH-specific insights.
	P. Liu et al. [20] (2024)	Hospital A: 85,193 patients with ECG or CXR and TTE randomly assigned to model development (n = 42,299), tuning (n = 17,182), or internal validation (n = 25,512). Hospital B: 16,736 patients with ECG or CXR and TTE as external validation set.	PASP > 50 mmHg estimated by TTE.	-AUCs for predicting ePAP: Internal validation: 0.8261 by ECG, 0.8525 by CXR, and 0.8644 by combination. External validation: 0.8348 by ECG, 0.8605 by CXR, and 0.8734 by combination. -Sensitivity and specificities were in the range of 71–86%, NPV in the range of 97–99%, and PPV in the range of 18–24%. -Combined ECG and CXR model predicted future risk of LV dysfunction and 8-year cardiovascular mortality [HR 4.51 (95% CI: 3.54–5.76), and 6.08 (4.66–7.95), respectively].	-PH is not diagnosed based on RHC. -Limited external validation.
**CXR**	Kusunose et al. [25] (2020)	-900 consecutive patients with suspected PH -A separate dataset of 90 consecutive patients to compare AI model with manual CXR measurements and human eyeball assessments -An independent external validation group of 55 patients with suspected PH.	RHC.	-AI model predicted PH with AUC ≈ 0.6 in first dataset. -AI algorithm showed higher AUC for detecting ePAP compared to manual CXR measurements and human eyeball assessments (0.71 vs. 0.60 and 0.63, all *p* < 0.05). -Patients with AI-predicted PH had a two-fold increased risk of heart failure hospitalization compared to those without (HR 1.9, 95% CI: 1.5–2.4).	-The inclusion of a high-risk population referred for RHC limits the generalizability of findings.
	Kusunose et al. [26] (2022)	142 high-risk patients for PAH (SSc or MCTD with SSc features) and normal resting TTE findings (52 had EIPH based on 6-MW stress echocardiography, of which 29 patients underwent exercise RHC).	Exercise RHC.	-AUC of the AI model was 0.71 (95% CI: 0.62–0.78). -Basic model (consisted of age, sex, blood pressure, and mean PAP at rest) was improved by AI model (adding the probability of PH calculated by the AI algorithm): AUC from 0.65 to 0.74; *p* = 0.046.	-Not all patients were confirmed by exercise RHC. -Small male sample size and potential cohort bias limit the generalizability.
	Zou et al. [27] (2020)	762 patients (405 with PH, 357 healthy controls) from three institutes. Whole sample comprised 762 images (641 for training and 80 for internal test from hospital A and B, and 41 from hospital C for external test).	PASP ≥ 40 mmHg estimated by TTE.	-Compared three models using transfer learning with pretrained parameters from ImageNet. The best-performing model achieved AUCs of 0.970 in internal and 0.967 in external tests for classifying PH from normal CXRs, demonstrating potential for initial PH screening, and showed good performance in predicting exact PASP values.	-PAP estimated by TTE. -Significant mean age difference between PH-positive and PH-negative groups.
	Han et al. [28] (2024)	-1174 CHD radiographs (including 142 PAH-CHD) and 2081 non-CHD radiographs. -A subset of 165 CHD [including 19 (11.52%) PAH-CHD] and 165 non-CHD radiographs randomly selected to compare radiologists and AI models, as well as radiologists with and without AI assistance.	PAH-CHD status determined by echocardiography.	Applied transfer learning with pretrained parameters from ImageNet. The AI model identified CHD and PAH-CHD from CXRs with higher AUCs than radiologists. AI assistance significantly improved radiologists’ diagnostic accuracy for PAH-CHD (AUC increased from 0.639 to 0.705, *p* < 0.05).	-PAH is not diagnosed based on RHC. -Imbalanced dataset for PAH-CHD.
	Imai et al. [29] (2024)	259 CXR images from 145 patients with PAH and 260 CXR images from 260 control patients. Divided into training (418 images) and test sets (101 images).	Confirmation of PAH by RHC.	AI model achieved a high AUC of 0.988 for detecting PAH, outperforming experienced doctors.	-Lack of external validation.
	Kıvrak et al. [30] (2022)	4497 CXRs from 2005 consecutive patients. 2145 CXRs from a publicly available dataset as the healthy group.	Echocardiography.	Compared to visual assessment and classification based on established guidelines by four physicians blinded to patients’ medical history, the best-performing model achieved an accuracy of 86.14% in classifying six categories (five PH types and non-PH). It reached an AUC of 0.945 in distinguishing between PH and non-PH cases.	-Unbalanced dataset in some categories. -Limited generalizability.
**Echocardiography**	Zhang et al. [34] (2018)	584 studies of 104 patients, and 2487 studies of 2180 controls.	Referring diagnosis. Confirmed if taking PAH-specific medications.	Detected PH with AUC of 0.85.	-Lack of external validation. -Relatively unbalanced dataset.
	Liao et Al. [35] (2023)	346 patients with suspected PH (240 PH patients and 106 non-PH) divided into training and internal validation sets: 275 patients (8:2 ratio), and external validation set: 71 patients.	RHC.	ML model outperformed traditional echocardiographic assessments [AUC 0.945 (95% CI: 0.917−0.974) in the internal validation (vs. 0.892, *p* = 0.027), and 0.950 in the external validation (95% CI: 0.897−1.000)].	-Limited generalizability.
	Ragnarsdottir et al. [36] (2024)	Training and validation set: 936 TTE videos from 192 newborns, held-out test set: 375 TTE videos from 78 newborns.	Visual evaluation by a senior pediatric cardiologist.	The spatio-temporal multi-view DLM achieved an F1-score of 0.84 for severity prediction and 0.92 for binary PH detection.	-Limited generalizability. -PH diagnosis and severity based on TTE.
	Anand et al. [37] (2024)	7853 patients (64% training set, 16% validation set, and 20% testing set).	RHC.	Predicted PH with AUC 0.83, sensitivity 88%, specificity 54%, without reliance on tricuspid regurgitation velocity.	-Limited generalizability (high prevalence of PH in the cohort). -Potential overfitting (performance drop between the training and testing sets).
	Hirata et al. [38] (2024)	885 patients (720 patients in derivation set divided into training and test sets in a 9:1 ratio, and 165 patients in validation set).	RHC (classification into non-PH, precapillary PH, and postcapillary PH).	Accurately classified PH subtypes with higher accuracy than guidelines in the derivation cohort (59.4% vs. 51.6%).	-Limited feature set included. -Limited accuracy in validation dataset.
	Li et al. [39] (2021)	151 CTEPH patients.	RHC.	Predicted adverse outcomes with HR 1.576.	-Limited generalizability.
	Fortmeier et al. [40] (2022)	116 patients for derivation, 142 patients for external validation.	RHC.	Predicted mPAP with high correlation to RHC (R = 0.96); mPAP > 29.9 mmHg associated with higher 2-year mortality.	-Limited external validation. -Some parameters sensitive to cardiac and volume status changes at the time of measurement.
	Murayama et al. [41] (2024)	85 patients (73 PH cases).	-RHC (detecting precapillary PH). -RVEF assessed by CMR.	-Accurately estimated RVEF with mean absolute error of 7.67%. -Comparable AUC for RV dysfunction to expert sonographers.	-Some parameters may have changed at the time of each assessment.
	Melzig et al. [46] (2019)	70 patients with suspected PH.	RHC.	-Predicted PH with AUC 0.90 for CTPA-derived MPA volumes, 0.92 for echocardiographic PASP, and 0.94 for predicted mPAP using combination of these parameters. -Predicted mPAP > 25.8 mmHg, identified PH with 86% sensitivity, 93% specificity, 95% PPV, and 81% NPV.	-Limited generalizability.
**CT**	Nardelli et al. [44] (2018)	-Trained and assessed on 21 non-contrast chest CT scans from the COPDGene study (compared with human observers). - Further tested on 33 contrast CT images (18 CTEPH patients, 15 controls).	CTEPH diagnosed by a panel of experts based on hemodynamics and imaging characteristics.	Achieved 94% classification accuracy (86.9% for CTEPH and 91.7% for controls), outperforming previous models.	-Small dataset. -Lack of external validation.
	Rahaghi et al. [45] (2021)	42 PAH, 12 ePH, 37 controls.	RHC.	Identified and quantified key vascular changes, vascular volume and manifestations of pruning and vascular tortuosity, using CT imaging in PAH and ePH.	-Small dataset. -Reliance on specific CT qualities. -Time gap between imaging and hemodynamic data.
	Zhang et al. [47] (2023)	55 PH patients (80% training data set and 20% independent testing data set).	RHC.	High correlation between AI-extracted CT features and mPAP (ICC > 0.9); AUC of 0.91 for mPAP classification.	-Limited generalizability. -Potential overfitting (limited training data).
	Charters et al. [48] (2022)	202 patients with suspected PH, split randomly into derivation cohort (n = 100), and validation cohort (n = 102).	RHC.	The automated RV/LV ratio achieved an AUC of 0.752 (95% CI: 0.677–0.827, *p* < 0.001) for predicting PH.	-Limited generalizability. -Time gap between imaging and hemodynamic data.
	Zhao et al. [49] (2024)	300 cases from 2 cohorts: Cohort 1: 88 cases of CTEPH, 80 cases of non-CTEPH lung diseases, and 8 healthy controls. Cohort 2: 124 healthy controls.	CTEPH defined via clinical and imaging criteria.	The model achieved better diagnostic performance (AUC: 0.807) compared to V/Q scan-based methods (AUC: 0.670) and NCCT methods without AI integration (AUC: 0.760) for CTEPH detection.	-Limited generalizability. -Unbalanced data.
	Gawlitza et al. [50] (2024)	127 preoperative CTEPH patients split into training and testing (3-folded random sampling for testing).	RHC.	AUC of 0.82 for mPAP prediction; identified key features like RV/LV ratio and mosaic attenuation patterns.	-Lack of external validation. -Manual feature selection. -Unbalanced data.
	Mamalakis et al. [51] (2023)	Primary “seen” cohort (training and validation): 84 patients with PH, “unseen” dataset (external validation): 19 patients with diverse pulmonary conditions.	-PH diagnosis standard not specified. - CT lung Pattern classification by 2 radiologists.	-The best-performing model achieved a classification accuracy of 93.69% and a Jaccard score of 91.83% in the “unseen” cohort. -Analysis revealed high confidence in detecting emphysema and honeycomb patterns but challenges with ground-glass reticulation and healthy patterns.	-Small sample size. -Unbalanced disease classes. -High uncertainty in specific classes.
	Dwivedi et al. [52] (2024)	Training cohort (n = 275), and test cohort (n = 246).	IPAH and PH-LD diagnosed via clinical and imaging criteria.	AI-quantified fibrosis was independently associated with increased mortality (HR: 1.01, *p* = 0.04), and improved survival prediction when combined with radiologic scoring (C-index: 0.67 vs. 0.61, *p* < 0.001).	-Unbalanced data. -Limited generalizability.
	Shikhare et al. [53] (2022)	99 operated CTEPH cases.	CTEPH diagnosis by a multidisciplinary team of specialists according to imaging and clinical data.	RV/LV ratio independently predicted prolonged ICU stay post-surgery (OR = 3.79, 95% CI 1.1–13.06, *p* = 0.035) and correlated with disease severity.	-The pretrained model was not trained on an internal cohort. -Lack of external validation. -ML tool failed in more severe cases.
**CMR**	Duan et al. [54] (2019)	-PH dataset: 649 CMR volumes -U.K. Digital Heart Project Dataset: 1831 healthy CMRs.	Not specified.	-The AI pipeline achieved segmentation accuracy comparable to manual methods.	-Lack of external validation. -Computationally expensive shape refinement.
	Attard et al. [55] (2019)	312 PH subjects (182 with metabolomics) and 1985 healthy volunteers.	Not specified.	The model demonstrated that RV wall stress, quantified through AI-assisted biomechanical modeling, was an independent predictor of survival in PH patients (hazard ratio 1.27, *p* = 0.04).	-Lack of external validation. -Heterogeneous patient cohort. -Time gap between tests.
	Ma et al. [56] (2021)	30 PH patients.	Not specified.	DL-based RV segmentation using YOLO and CNN achieved high accuracy (Dice coefficient: 0.89, accuracy: 99.4%) and identified PH-related morphological features.	-Small sample size. -Lack of external validation.
	Lungu et al. [57] (2016)	72 patients with suspected PH (57 confirmed with PH, and 15 classified as non-PH).	RHC.	The model achieved a diagnostic accuracy of 92% (sensitivity: 97%, specificity: 73%). Adding computational and image-based metrics significantly enhanced diagnostic accuracy (AUC from 0.89 for standalone computational metrics to 0.91 for the combined model).	-Imbalanced data set. -Lack of external validation.
	Swift et al. [58] (2021)	220 patients with suspected PH (150 diagnosed with PAH, 70 classified as “No PH”).	RHC.	The model achieved a high diagnostic accuracy: AUC of 0.90 for short-axis cine images and 0.86 for four-chamber cine images. For differentiating IPAH from controls: AUC of 0.97 for short-axis cine images and 0.95 for four-chamber cine images.	-Limited scope (2 CMR views). -Lack of external validation.
	Priya et al. [59] (2021)	42 PH patients and 30 controls split into training and test sets.	RHC.	The model using texture features from CMR achieved high accuracy in differentiating PH patients from controls, with the MLP model performing best (AUC: 0.862, accuracy: 78%).	-Lack of external validation. -Focused exclusively on LV texture features.
	Priya et al. [60] (2021)	42 PH, 40 controls.	RHC.	Radiomics-based ML using CMR with the DAFIT technique achieved high diagnostic accuracy (AUC: 0.958) in differentiating PH patients from controls.	-Small sample size. -Lack of external validation. -Technical complexity.
	Wang et al. [61] (2024)	-Primary dataset 7900 (including 1250 controls, 134 PAH, and 6516 other CVDs) -External test dataset 1819 (including 403 controls, 66 PAH, and 1350 other CVDs).	RHC.	AI-enabled CMR approach achieved 99.8% AUC in detecting PAH by combining cine and late gadolinium enhancement imaging, outperforming expert cardiologists, and demonstrating robust external validation across multiple centers.	-Unbalanced dataset.
	Dawes et al. [62] (2017)	256 PH patients.	RHC.	The model using 3D RV motion patterns from CMR improved survival prediction in PH patients, increasing the AUC from 0.60 to 0.73 (*p* < 0.001).	-Lack of external validation. -Limited generalizability.
	Alandejani et al. [63] (2022)	-Training dataset: 365 patients/447 studies (285 patients from the ASPIRE registry, 29 healthy and 37 with myocardial infarction from one center, and 14 healthy participants from another center.) -Test dataset: 400 patients from the ASPIRE registry for clinical validation -A cohort of 36 patients from the RESPIRE study for interstudy repeatability assessment - 3795 patients from the ASPIRE registry for quality control evaluation.	RHC.	The model for automated RA area measurement using CMR demonstrated higher repeatability (ICC: 0.91–0.95) than manual methods, showed moderate correlation with mean RA pressure (*r* = 0.64), and was equally predictive of mortality as manual measurements (HR: 1.02, *p* < 0.01).	-Lack of external validation.
	Alabed et al. [64] (2022)	Training dataset (539 patients from two centers with various cardiac conditions), testing dataset (3487 patients from the ASPIRE registry and external validation with 40 multivendor studies from 32 centers).	RHC.	The model for automated CMR analysis demonstrated strong correlations with RHC parameters (mPAP *r* = 0.56, PVR *r* = 0.62), high repeatability (ICC: 0.90–0.99), and accurate mortality prediction (HR 1.4 for RV mass, *p* < 0.001).	-Small external validation set.
	Alabed, Alandejani, et al. [65] (2022)	723 PAH patients (training cohort 516 patients, validation cohort 207 patients).	RHC.	The model using time-resolved CMR features significantly improved 1-year mortality prediction in treatment-naïve PAH patients, increasing the c-index from 0.71 (REVEAL score) to 0.83 (*p* < 0.001) by identifying dynamic interventricular septal and LV changes linked to survival.	-Lack of external validation.

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
