# Peer review of "A Comprehensive Review of Artificial Intelligence (AI) Applications in Pulmonary Hypertension (PH)"

_medicina, 2025, doi:10.3390/medicina61010085_

Round 1
Reviewer 1 Report
Comments and Suggestions for Authors
Thank you for inviting me to review this paper on hot topic use of AI in pulmonary hypertension.
I have some important suggestions:
1. please add chapter future directions with your opinion about impact of implementation of ai in real life practice in pulmonary hypertension
2. Discussion is extremely short please rewrite it completely
3. tables and figures are missed in this version, please add all of them!
Author Response
Responses to Reviewer 1 comments
Comment 1. Please add chapter future directions with your opinion about impact of implementation of AI in real life practice in pulmonary hypertension.:
Authors: Thank you for your suggestion. We have added a dedicated ‘Future Directions’ paragraph under the Discussion section on pages 26-28, where we provide our perspective on the potential impact of implementing AI in clinical practice for pulmonary hypertension.
Comment 2. Discussion is extremely short please rewrite it completely:
Authors: The discussion has been entirely rewritten to expand on key findings and integrate new perspectives.
Comment 3. Tables and figures are missed in this version, please add all of them.
Authors: We apologize for the omission of information in the previous version of the manuscript. In the revised version, tables and figures summarizing key findings have been incorporated.
Reviewer 2 Report
Comments and Suggestions for Authors
I thank the Editor for allowing me an opportunity to review the manuscript ‘A Comprehensive Review of Artificial Intelligence Applications in Pulmonary Hypertension (PH).’
Sogol et al. sought to highlight the significant promise of AI in revolutionizing PH management to im- 23 prove patient outcomes by means of a narrative review.
I read the investigation with interest. The manuscript flows well overall and is easy to read. Please consider the points below, in no specific order of importance.
• I noticed that the manuscript moves straight from the ‘Introduction’ section to the ‘Results’ section. As such, this review is very stingy in relation to the available databases and the rigor/soundness of the underlying methodology. I expected it to be more informative and detailed.
• This Reviewer would strongly advise the authors to implement a new ‘Materials and Methods’ section where they describe the methodology employed to conduct the literature search and subsequent article retrieval.
• Even if this is not a systematic review, I still believe that a PICO/SPIDER framework was developed and I would like to know if the SANRA criteria for narrative reviews were abided by.
• Furthermore, I am interested in understanding inclusion/exclusion criteria for the retrieved articles as well as key terms employed to conduct the literature search, if snowballing was performed and also who and how many investigators were involved in each phase of this process.
• Additionally, I am entirely unsure about the quality assessment of the retrieved studies – was this performed? If so, how? I believe that even if this is a narrative review, the assessment of the Risk of Bias with RoB2/ROBINS-I should have been performed. Please elaborate on this and include it within your main manuscript and/or your Supplementary material.
• Furthermore, the authors discuss a huge number of studies in their review, which is good in principle. Nevertheless, I believe that it would be useful (and it would also make the average reader happy!) to further enrich the quality of the manuscript by providing Tables within your main manuscript (ideally, one for each of the sub-areas delineated in paragraphs 2.1-2.3 and sub-sub-paragraphs, respectively, where AI can be implemented in the setting of PH) in which you summarize in detail key info (First author, publication year, setting, no. of patients, disease(s), key findings) from each of the retrieved study in a schematic, straightforward way.
• One other aspect in which AI shows promise to revolutionize care is the field of critical illness (i.e., cardiogenic shock) which can – to some extent – be concomitant in this patient population. I would advise the authors to mention this aspect within their ‘Introduction’ or ‘Discussion’ section aided by a summary and quoting of PMID: 39049432 and PMID: 38783580.
• I believe a comprehensive Figure should be added to the manuscript which provides a visual representation of all the different areas where AI can be implemented and be also of aid in the setting of PH diagnosis and management.
Author Response
Responses to Reviewer 2 comments
Sogol et al. sought to highlight the significant promise of AI in revolutionizing PH management to improve patient outcomes by means of a narrative review.
I read the investigation with interest. The manuscript flows well overall and is easy to read. Please consider the points below, in no specific order of importance.
 
Comment 1. I noticed that the manuscript moves straight from the ‘Introduction’ section to the ‘Results’ section. As such, this review is very stingy in relation to the available databases and the rigor/soundness of the underlying methodology. I expected it to be more informative and detailed.
Comment 2. This Reviewer would strongly advise the authors to implement a new ‘Materials and Methods’ section where they describe the methodology employed to conduct the literature search and subsequent article retrieval. 
Comment 3. Furthermore, I am interested in understanding inclusion/exclusion criteria for the retrieved articles as well as key terms employed to conduct the literature search, if snowballing was performed and also who and how many investigators were involved in each phase of this process.
Authors response to comments 1, 2, and 3: To further address in detail each of these points highlighted by the reviewer, we have added a ‘Materials and Methods’ section at the beginning of page 2 that describes the databases used, key terms employed, and a sentence that states that snowballing techniques were not employed.
Comment 4. Even if this is not a systematic review, I still believe that a PICO/SPIDER framework was developed, and I would like to know if the SANRA criteria for narrative reviews were abided by. 
Authors: We utilized a thematic framework to structure the review and ensured familiarity with the SANRA scale. Following a thorough review of the feedback we applied this scale to assess the manuscript, assigning it a score of 10 out of 12, which reflects its adherence to high standards for a narrative review. The SANRA scale has been included as Table A2 (Appendix A).
Comment 5. Additionally, I am entirely unsure about the quality assessment of the retrieved studies – was this performed? If so, how? I believe that even if this is a narrative review, the assessment of the Risk of Bias with RoB2/ROBINS-I should have been performed. Please elaborate on this and include it within your main manuscript and/or your Supplementary material.
Authors: To assess the risk of bias in the retrieved studies, we used PROBAST (Prediction model Risk of Bias Assessment Tool). A table summarizing the PROBAST assessment has been added to the manuscript (Table A1, Appendix A). Given that the studies we have reviewed in this article focus on diagnostic and prognostic models, we believe PROBAST is more applicable than RoB2 or ROBINS-I.
Comment 6. Furthermore, the authors discuss a huge number of studies in their review, which is good in principle. Nevertheless, I believe that it would be useful (and it would also make the average reader happy!) to further enrich the quality of the manuscript by providing Tables within your main manuscript (ideally, one for each of the sub-areas delineated in paragraphs 2.1-2.3 and sub-sub-paragraphs, respectively, where AI can be implemented in the setting of PH) in which you summarize in detail key info (First author, publication year, setting, no. of patients, disease(s), key findings) from each of the retrieved study in a schematic, straightforward way.
Authors: Thank you for this insightful suggestion. We have added Table 1 in this version of the manuscript, which describes and summarizes each of the studies evaluating AI in pulmonary hypertension.
Comment 7. One other aspect in which AI shows promise to revolutionize care is the field of critical illness (i.e., cardiogenic shock) which can – to some extent – be concomitant in this patient population. I would advise the authors to mention this aspect within their ‘Introduction’ or ‘Discussion’ section aided by a summary and quoting of PMID: 39049432 and PMID: 38783580.
Authors: Thank you for this suggestion, we have included these references in the discussion section.
Comment 8. I believe a comprehensive Figure should be added to the manuscript which provides a visual representation of all the different areas where AI can be implemented and be also of aid in the setting of PH diagnosis and management.
Authors: Thank you for this suggestion. In the new version of the manuscript, we have added Figure 2, which depicts in a comprehensive manner the role of AI in the setting of PH diagnosis and management.
Round 2
Reviewer 1 Report
Comments and Suggestions for Authors
accept as it
Reviewer 2 Report
Comments and Suggestions for Authors
I thank the Authors for thoroughly revising their manuscript in response to this expert Reviewer's suggestions.
I believe that their efforts have been beneficial and that the revised version of this manuscript has now significantly improved - I am genuinely persuaded that this is now an outstanding manuscript in the current form - very comprehensive and well-organized. Congratulations!
I only have a MINOR suggestion for the authors to consider which can though be addressed in subsequent draft/proof stage - I would remove the two abbreviations (i.e., AI and PH) from the title, to increase its readability.